# Effect of SNPs in the Promoter Region on the Expression of Cytochrome P450 2E1 (CYP2E1) in Pig Liver

**DOI:** 10.3390/ani14081163

**Published:** 2024-04-12

**Authors:** Holly Archer, Riani A. N. Soares, Mohsen Jafarikia, Brandon N. Lillie, Flavio Schenkel, E. James Squires

**Affiliations:** 1Department of Animal Biosciences, University of Guelph, Guelph, ON N1G 2W1, Canadamjafarik@uoguelph.ca (M.J.); schenkel@uoguelph.ca (F.S.); 2Canadian Centre for Swine Improvement Inc., Ottawa, ON K1A 0C6, Canada; 3Department of Pathobiology, Ontario Veterinary College, University of Guelph, Guelph, ON N1G 2W1, Canada; blillie@uoguelph.ca

**Keywords:** gene expression, promoter region, single-nucleotide polymorphism

## Abstract

**Simple Summary:**

Cytochrome P450 2E1 (*CYP2E1*) is a key enzyme involved in the hepatic metabolism of the major boar taint compound, skatole. Here, we identified single-nucleotide polymorphisms (SNPs) in the promoter region of the *CYP2E1* gene that were associated with *CYP2E1* mRNA expression, but not with CYP2E1 protein expression. This demonstrates the role of promoter SNPs in the expression of *CYP2E1* mRNA and suggests that factors that regulate the translation of *CYP2E1* mRNA may also be important in affecting skatole metabolism.

**Abstract:**

Boar taint, an unfavorable odor in the meat of intact male pigs, is caused primarily by the accumulation of two compounds: androstenone and skatole. This multifactorial trait is regulated by numerous dietary, management and genetic factors. At the mechanistic level, there are many genes known to be involved in boar taint metabolism. Cytochrome P450 2E1 (CYP2E1) impacts boar taint through the phase I metabolism of skatole. The aim of this study was to identify single-nucleotide polymorphisms (SNPs) within the *CYP2E1* gene promoter and explore their relationship with the expression of *CYP2E1* mRNA and protein. Sequencing of the promoter region using pools of genomic DNA identified seven promoter region SNPs at −159, −586, −1693, −1806, −2322, −2369 and −2514 bp upstream of the ATG start site. Genomic DNA was obtained from 65 boars from the three major swine breeds: Duroc, Landrace and Yorkshire, and individual animals were genotyped for the identified SNPs. RNA was isolated from liver tissue and quantitative PCR was performed to measure *CYP2E1* gene expression, while levels of CYP2E1 protein in liver were measured by Western blotting. Significant within-breed variation in CYP2E1 protein and mRNA expression was observed, indicating significant differences in gene expression among individuals. However, levels of *CYP2E1* mRNA and protein were not significantly correlated. Two SNPs within the promoter were significantly associated with *CYP2E1* mRNA expression, but not with protein expression. This study provides evidence of additional mutations affecting the gene expression of *CYP2E1* and suggests that factors that affect the differences in translation of *CYP2E1* mRNA may also be important in affecting skatole metabolism.

## 1. Introduction

In swine production, male pigs are routinely castrated in many countries to prevent the development of an unpleasant odor or flavor in the meat of entire male pigs, known as boar taint. This is caused by the accumulation of androstenone (5α-androst-16-ene-3-one) or skatole (3-methylindole) in the adipose tissue, which are produced during steroidogenesis in the testes or from the bacterial degradation of tryptophan in the gastrointestinal tract, respectively [1,2]. The accumulation of these compounds is dependent on their rate of synthesis as well as their rate of clearance (reviewed in [3]). The amount of boar taint can be affected by various factors, including environment and management practices, sexual maturity, nutrition and genetics. There are distinct breed differences in the amount of boar taint, with higher levels of skatole and androstenone and boar odor found in Large White boars compared to Pietran and higher levels of androstenone in Duroc compared to Landrace boars. Overall, higher levels of boar taint compounds are found in dam compared to sire lines (reviewed in [4]). It may be possible to decrease the occurrence of boar taint by manipulating factors that reduce the synthesis or increase the metabolism of androstenone or skatole.

The metabolism of androstenone and skatole occurs primarily in the liver in a two-phase process. Phase I involves a series of oxidation, reduction or hydrolysis reactions, which facilitate the addition of a hydroxyl group to the compound [3]. Next, the compound undergoes phase II conjugation reactions, which involve the transfer of a sulfonate group or glucuronic acid, from the donor molecule 3′-phosphoadenosine 5′-phosphosulfate (PAPS) or uridine diphosphate glucuronic acid (UDPGA), respectively, to the hydroxyl group of the acceptor molecule [5]. This two-phase process acts to render the compound inactive and more water soluble to aid its subsequent clearance [6]. The oxidation reactions of phase I metabolism are mediated by a family of heme-thiolate enzymes known as cytochrome P450s [7]. Of the various cytochrome P450 enzymes, cytochrome P4502E1 (*CYP2E1*) has been identified as an important regulator of skatole metabolism [8,9] and can be considered a key candidate gene affecting boar taint from skatole.

Several studies have implicated CYP2E1 as an important enzyme in skatole metabolism by demonstrating a negative correlation between *CYP2E1* expression and the concentration of skatole in the adipose tissue, depending on the breed. For example, Whittington et al. [10] observed low hepatic *CYP2E1* gene expression in Meishan X Large White pigs that exhibited high skatole concentrations in the fat. Additionally, Squires and Lundström [8] demonstrated a negative correlation between CYP2E1 liver protein levels and skatole concentrations in the backfat of a cross of European Wild Pig x Swedish Yorkshire domestic boars. These results suggest that *CYP2E1* expression is a major factor influencing the accumulation of skatole in the adipose tissue in a breed-dependent manner, which could potentially be manipulated to reduce the occurrence of boar taint. 

Several SNPs have been identified in the promoter region of *CYP2E1*, but only one of these (at −586 ATG) was associated with skatole levels [11,12]. However, the effect of this SNP on the expression of *CYP2E1* has not been investigated. The purpose of this study was to identify additional mutations in the promoter region of *CYP2E1* and to determine their potential effect on the expression of *CYP2E1* mRNA and protein. 

## 2. Material and Methods

### 2.1. SNP Discovery

Genomic DNA was extracted using the GenElute Genomic DNA miniprep kit (Sigma, Oakville, ON, Canada) from 10 boars with high boar taint and 10 boars with low boar taint from purebred Duroc, Yorkshire and Landrace, which were selected based on a threshold value of 0.5 µg/g and 0.25 µg/g for fat androstenone and skatole concentrations, respectively. Equal aliquots (4 μg) from each of the 10 high- and 10 low-taint boars in each breed were pooled, and the promoter sequence of *CYP2E1* (clone AJ697882.1, [11]) was amplified by PCR using the following primers: CYP2E1-74-F 5′-CAAGCTGACTGTCCCTTTGG-3′, CYP2E1-674-F 5′-TCTGCTGTTCTCCCAGGACT-3′; CYP2E1-1263-F, 5′-GCTTGAAAATGTCATTACTTCCA-3′ with reverse primer CYP2E1-2987-R 5′-GGGGATCTTTCAATGTGTGG-3′. PCR conditions were 94 °C for 3 min, 35 cycles of annealing temperature of 60 °C for 1 min, extension temperature of 72 °C for 1 min, followed by 72 °C for 10 min. The quality of the PCR products was checked by agarose gel electrophoresis and the DNA was then sequenced using the same primers. The sequencing chromatograms were checked using Sequencher 4.9 software (Gene Codes Corporation, Ann Arbor, MI, USA) for the appearance of double peaks, which indicated the presence of a potential polymorphism at that location in the gene sequence. Seven SNPs were subsequently confirmed in the promoter region of *CYP2E1* at −159, −586, −1693, −1806, −2322, −2369 and −2514 bp upstream of the ATG site.

### 2.2. Animals

The sample population for *CYP2E1* mRNA and protein expression analyses consisted of purebred Duroc, Yorkshire and Landrace boars from a total of 21 herds throughout Canada. Due to the availability of tissue samples and genotypic data, 65 boars (20 purebred Landrace, 25 Yorkshire and 20 Duroc boars) sourced from swine breeders in Ontario, Canada, were included in this study and used for the quantification of *CYP2E1* mRNA and protein. Boars were kept under standard commercial housing conditions and fed a commercial corn/soy-based finisher ration. They were humanely slaughtered at a weight of 109.3 ± 0.4 kg and 150–160 days of age at the University of Guelph Meat Lab (Canadian Food Inspection Agency Establishment 183). All animals were used in accordance with the guidelines of the Canadian Council of Animal Care and the University of Guelph Animal Care Policy (Animal Utilization Protocol number 3723 approved 5 July 2021).

### 2.3. Genotyping

Genomic DNA was extracted from the 65 boars (20 purebred Landrace, 25 Yorkshire and 20 Duroc boars) used in this study and samples were genotyped for these SNPs using the Sequenom Mass Array System (University Health Network, Toronto, ON, Canada). Imputation of missing genotypes (18.54% were missing) was carried out on genotypic data using family + population imputation with the software package FImpute 2.2 [13]. Genotype output was recoded to phased genotypes as follows: 0: A1A1; 1: A1A2 and A2A1; 2: A2A2; 5: missing. After the imputation, there were no genotypes missing.

### 2.4. Isolation of Total RNA

Liver samples were obtained from the 65 animals used in this study. Liver tissue was removed immediately following exsanguination, frozen in liquid nitrogen and stored at −70 °C prior to use. RNA was isolated by incubating liver samples for 5 min on ice in 1 mL of Tri-Reagent (Sigma, Oakville, ON, Canada) per 100 mg sample. Next, 100 µL of 1-bromo-3-chloropropane was added to the homogenate and the samples were vortexed and incubated at room temperature for 8 min. The samples were then centrifuged at 13,000× *g* for 15 min. After centrifugation, the aqueous phase containing the RNA was added to 400 µL of isopropanol, kept on ice for 5 min and then centrifuged at 13,000× *g* for 8 min to pellet the RNA. The supernatant was removed and 1 mL of 75% ethanol in diethyl pyrocarbonate (DEPC) (Sigma, Oakville, ON, Canada) was added. After centrifugation, the RNA pellet was air-dried, dissolved in 50 µL of DEPC water and stored at −70 °C until use.

### 2.5. cDNA Synthesis and Real-Time PCR

cDNA was synthesized using a SuperScript ^™^ II Reverse Transcriptase kit (Invitrogen Corp, Burlington, ON, Canada), and the cDNA samples were stored at −20 °C until use. Following reverse transcription, real-time qPCR was carried out to determine the expression level of *CYP2E1* in each liver sample following procedures previously described [14] with modifications. Briefly, the reaction was conducted in 96-well plates with PCR reagent mixtures as follows: 5 µL Rotor-Gene SYBR Green PCR Master Mix (Qiagen, Toronto, ON, Canada), 0.5 µL for both forward and reverse primer, 100 ng cDNA sample and 3.5 µL of H_2_O for a total reaction volume of 10 µL per well. The forward (5′-TCACTGTGTACCTGGGTTCG-3′) and reverse (5′AAAATGACCCCTTTGTCCTTGTG-3′) primers were designed using Primer-Blast (NCBI) in order to amplify a ~200 bp section of the *CYP2E1*-coding region. Cycling parameters consisted of 95 °C for 5 min followed by 45 cycles of [95 °C for 5 s, 60 °C for 10 s] and was carried out on the Rotor Gene 3000 Real-Time Thermal Cycler (Corbett Life Science, Concord, NSW, Australia). Each assay was conducted in duplicate, and relative fold expression was calculated using the 2^−ΔΔCT^ method [15], which was normalized to the internal β-actin reference. β-Actin was selected due to its similarity in abundance and amplification efficiency to *CYP2E1*. Melt curves were generated via 1 °C sequential temperature increases from 60 °C to 95 °C.

### 2.6. Western Blotting

Western blot analysis for CYP2E1 was performed on liver samples from the 65 boars using the methods previously described [16] with modifications. Briefly, using 10% sodium dodecyl sulfate–polyacrylamide gel electrophoresis (SDS-PAGE), 5 µg of protein from boar liver homogenate was loaded per well in duplicate. Following separation, the proteins were trans-blotted to a polyvinylidene difluoride (PVDF) membrane by a semi-dry trans-blot. Membranes were incubated overnight at 4 °C in 5% (*w*/*v*) dried skim milk in phosphate-buffered saline (PBS) with 0.1% Tween 20 and then incubated with primary goat anti-rat CYP2E1 antibody (40,000 × dilution) (Daiichi Pure Chemicals Ltd., Tokyo, Japan). The specificity of this antibody for pig CYP2E1 has been reported previously [17]. The membranes were then incubated with a secondary rabbit anti-goat IgG antibody conjugated with horseradish peroxidase (8000 × dilution) (Sigma, St. Louis, MO, USA) in 5% skim milk in PBS. Protein bands were visualized by incubation with detection solution (50 µL of 68 mM p-coumaric acid in dimethyl sulfoxide (DMSO), 5 mL of 1.25 mM luminol in 0.1 M Tris, pH 8.5, and 15 µL of 3% hydrogen peroxide) for 1 min and quantified using a ChemiDoc MP imaging system (Bio-Rad, Mississauga, ON, Canada) with Image Lab 5.0 software.

### 2.7. Statistical Analysis

Allele frequencies were calculated for each SNP on the 65 genotyped animals using SAS PROC FREQ (Version 9.4). Gene expression (fold change) and protein expression (relative band intensity) in the liver were regressed on the number of copies of the mutant alternate allele (i.e., 0, 1, 2) to estimate the allele substitution effect using SAS PROC GLM (Version 9.4) and a univariate model, which included breed and weight as fixed effects, as follows:$$Y_{jk}=\sum_{i}^{n}b_{i}G_{ij}+\beta w_{j}+B_{k}+e_{jk}$$
where $Y_{jk}$ is the *CYP2E1* gene expression (fold change) or protein expression (relative band intensity) in the liver of animal *j* of breed *k*, $G_{ij}$ is the recoded genotype (0, 1 or 2) of SNP *i* of animal *j*, $w_{j}$ is the weight of animal *j*, $B_{k}$ is the breed *k*, $e_{jk}$ = is the residual random term for animal *j*, $b_{i}$ = is the coefficient of the fixed linear regression on the recoded genotypes (allele substitution effect) for SNP *i*, β = is the coefficient of the fixed linear regression on the weight of animal *j*, and *n* is the number of SNPs. 

Pearson’s correlation coefficient (r) between recoded genotypes was used as an approximated measure of linkage disequilibrium among SNPs [18]. One SNP of any pair of SNPs with a high correlation between recoded genotypes |r| ≥ |0.97| was removed from the analysis. After that, partial correlations between the remaining SNPs were estimated, and one SNP of any pair of SNPs with high partial correlation |r| ≥ |0.97| was removed from the analysis.

The association between levels of CYP2E1 protein expression and of *CYP2E1* mRNA expression was determined by linear regression of levels of CYP2E1 protein expression on *CYP2E1* mRNA fold change and carcass weight of the pigs within and across breeds using PROC GLM in SAS (Version 9.4). The across-breed analysis included breed effect in the model.

## 3. Results

### 3.1. CYP2E1 mRNA and Protein Expression

Significant within-breed variation was observed in *CYP2E1* mRNA expression (Table 1) in terms of fold change, with a range from 0.04 to 39.67 across breeds. However, there was no significant difference among breed groups in mRNA expression. 

Western blot results revealed that relative CYP2E1 protein expression ranged from 0 to 1513, with some significant differences (*p* < 0.05) between breed groups (Table 1). Protein levels in the Yorkshire breed were lower than in the Landrace and Duroc breed groups. The Landrace and Duroc breed groups were not different from each other. Test for normality showed that CYP2E1 protein expression was normally distributed within and across breeds (*p* > 0.05).

Carcass weight had a significant effect on *CYP2E1* mRNA fold change (*p* < 0.05), but not on relative CYP2E1 protein expression (*p* > 0.05). Thus, the final regression linear model to evaluate the association between levels of CYP2E1 protein expression and of *CYP2E1* mRNA fold change did not include carcass weight. Levels of CYP2E1 protein were not significantly associated with levels of *CYP2E1* mRNA fold change both within and across breeds (non-significant regression coefficients (*p*> 0.05); Figure 1).

### 3.2. SNP Effect on mRNA Expression 

A summary of *CYP2E1* promoter SNP genotype statistics is presented in Table 2. SNPs were identified at −159, −586, −1693, −1806, −2322, −2369 and −2514 bp upstream of the ATG site. All SNPs had a minor allele frequency >0.05 and ranged from 0.154 to 0.423. 

The results of the association analyses between SNP genotypes and mRNA expression (fold change) are presented in Table 3. A combined SNP effect model was run to account for potential linkage between the SNPs. As mentioned before, to prevent high collinearity, one SNP of any pair of SNPs with a high correlation between recoded genotypes (0, 1 or 2) (|r| ≥ |0.97|) was removed from the analysis. The three remaining SNPs had a partial correlation between each other from 0.12 to −0.71. SNP S90 (−2390 ATG) had the highest numerical impact on gene expression, showing a significant allele substitution effect on fold change of −3.57 ± 1.57 (*p* = 0.03), followed by SNP S102 (−1693 ATG) with an allele substitution effect of −3.38 ± 1.28 fold change (*p* = 0.01). SNP S35 (−159 ATG) had no significant effect on fold change (*p*= 0.55).

## 4. Discussion

Cytochrome P4502E1 is known to metabolize a wide variety of low-molecular-weight xenobiotics and is one of the main enzymes responsible for skatole metabolism in pigs (reviewed in [3]). When considering the mutations already identified within this gene, there have been relatively few mutations in the promoter region, while many have been outlined within the remaining region of *CYP2E1* gene [19]. Skinner et al. [11] identified four potential promoter region SNPs in pig *CYP2E1*, located at positions 1956 (C > G), 2108 (A > G), 2115 (A > T) and 2412 (C > T) of the GenBank sequence AJ697882. However, their results indicated that three of these were fixed, and the sole remaining segregating SNP (AJ697882_2412, which is equivalent to −586 ATG) was found to have no significant association with skatole or indole measurements in their Large White/Meishan experimental cross, but it was significantly associated with skatole levels in a population of Danish commercial Landrace/Yorkshire/Duroc pigs. These results further emphasize the effect of breed on the levels of skatole in intact male pigs. Numerous studies have identified the *CYP2E1* promoter as highly influential in translation efficiency [20]. Therefore, the current study was designed to identify additional mutations in the promoter region of *CYP2E1* and to determine their potential effect on the expression of *CYP2E1* mRNA and protein.

### 4.1. Carcass Weight and CYP2E1 Expression

The significant effect of weight on *CYP2E1* mRNA expression found in this study agrees with previous research by Whittington et al. [10], who demonstrated a positive correlation between gene expression and weight. This association was also confirmed in a trial conducted by Zamaratskaia et al. [21]. In the current study, weight was not significantly different across breeds (Table 1), and *CYP2E1* mRNA expression was significantly lower in high-weight pigs regardless of breed. This weight-related decrease in *CYP2E1* expression may result in decreased metabolism of skatole and an increase in adipose tissue skatole content. The effect of weight on mRNA expression is possibly due to the relationship between weight and sexual maturity. It is likely that live weight and gender-specific variations in *CYP2E1* activity are related to testicular steroid concentration. 

### 4.2. SNP Discovery and Effects on CYP2E1 mRNA and Protein Expression 

Sequencing of the promoter region of *CYP2E1* using pools of genomic DNA from boars from the three breeds identified SNPs at −159, −586, −1693, −1806, −2322, −2369 and −2514 bp upstream of the ATG site. As the genotypes of many of these SNPs were highly correlated, only three SNPs were included in the association analysis, and two of them were significantly associated with the expression of *CYP2E1* mRNA (fold change). Two important regulatory regions within the *CYP2E1* promoter, binding sites for HNF-1 and HNF-4, are located at −115/−101 and −142/−131, respectively [22]. SNP S35 (at −159 ATG) was in close proximity to these important regions, but it was not significantly associated with *CYP2E1* mRNA expression in the present study. Liu and Gonzalez [23] showed that co-transfection with HNF1-α was effective in activating *CYP2E1* gene expression, which demonstrates the importance of HNF-1 binding sites to *CYP2E1* transcriptional efficiency. Gray and Squires [14] also identified the role of the nuclear receptor FXR in regulating *CYP2E1* expression. 

SNP S36 at −586 ATG was identified by Skinner et al. [11] and shown to be associated with boar taint from skatole. In the current study, SNP S36 was removed due to the high partial correlation (r > 0.975) with SNP S35. However, SNP S35 was found to not be significantly associated with *CYP2E1* expression. SNP S36 was later investigated by Morlein et al. [12] using two Duroc-sired crossbreeds, who reported significant associations between the C > T mutation at −586, in which the CC genotypes had higher skatole values when compared to the CT and TT genotypes. Zadinová et al. [24] also found a significant difference between the genotypes, with the TT genotype being associated with a lower indole level. 

Many times, the underlying assumption is that protein expression is correlated to mRNA expression [25], whereby mRNA concentrations are used as proxies for protein concentrations and, therefore, the corresponding enzymatic activity. However, levels of many bacterial and eukaryotic proteins are not strongly correlated to mRNA concentrations [26] and 60% of protein concentration variation cannot be explained by mRNA concentrations. This is because protein levels are regulated by a breadth of cellular processes, including transcription, mRNA processing, translation, localizations, modifications, binding and programmed deconstruction [25]. 

Limited studies have been carried out on the correlation between *CYP2E1* mRNA and protein expression in pigs. Unlike other inducible P450s, increases in liver CYP2E1 protein levels are not always accompanied by increased mRNA levels [27]. Often, CYP2E1 protein expression and corresponding activity are regulated at the post-transcriptional level [28] with protein stabilization being an important factor [20]. Kocarek et al. [20] identified *CYP2E1* expression regulation mechanisms in rats and noted significant hepatic post-transcriptional modifications. Specifically, upwards of 40% of *CYP2E1* mRNA was not involved in protein synthesis. Castration of pigs has been shown to result in increased *CYP2E1* mRNA without affecting CYP2E1 protein and activity levels [29]. 

Other factors, such as the expression of CYB5A, can also affect CYP2E1 enzyme activity [17]. Other isoforms of cytochrome P450, including CYP1A1, CYP2A19, CYP2C33, CYP2C49 and CYP3A, also metabolize skatole [17], so CYP2E1 expression would only be important in skatole metabolism in those boars that have high levels of CYP2E1 and lower levels of these other important cytochrome P450s. In addition, enhanced hepatic metabolism of skatole in vivo would only be important in those boars that produce high levels of skatole in the gut. Factors that affect the translation, stability and activity of CYP2E1 protein in pig livers need to be characterized to fully understand its role in skatole metabolism in pigs.

## 5. Conclusions

Understanding the genetic basis for differences in the metabolism of boar taint compounds may lead to effective marker-enhanced breeding programs in pigs. Here, we identified SNPs in the promoter of *CYP2E1*, a key candidate gene involved in skatole metabolism. Two of these SNPs were associated with the expression of *CYP2E1* mRNA, but not CYP2E1 protein, and levels of CYP2E1 protein were not correlated with levels of *CYP2E1* mRNA. These results demonstrate the role of promoter SNPs on the expression of *CYP2E1* mRNA and suggest that factors that affect the differences in translation of *CYP2E1* mRNA may also be important in affecting skatole metabolism.

## Figures and Tables

**Figure 1 animals-14-01163-f001:**
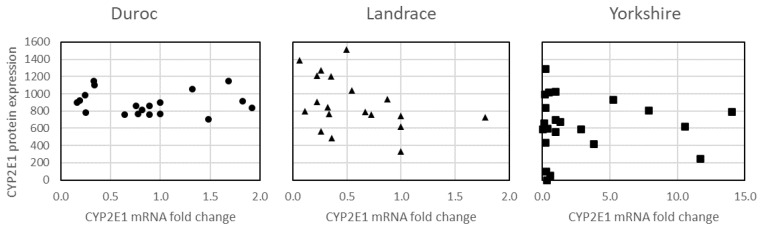
Plots of CYP2E1 protein expression versus mRNA fold change for three breeds of pigs displaying the lack of linear association between these parameters (non-significant regression coefficients; *p* > 0.05).

**Table 1 animals-14-01163-t001:** Carcass weight and expression of CYP2E1 protein and mRNA of boars used in this study.

Breed	Carcass Weight (kg)	mRNA by qPCR (Fold Change)	Relative CYP2E1 Protein Expression
Duroc	113.3 ± 4.35 ^a^	1.42 ± 1.19 ^a^	874.3 ± 68.7 ^a^
Landrace	112.4 ± 4.35 ^a^	1.30 ± 1.18 ^a^	956.3 ± 67.7 ^a^
Yorkshire	103.3 ± 3.89 ^a^	3.09 ± 1.03 ^a^	639.6 ± 59.4 ^b^

The values reported are derived from a sample size of 65 boars (20 Duroc, 20 Landrace and 25 Yorkshire). Values are presented as least squares means ± SE. In the same column, different letters indicate means that are significantly different (*p* < 0.05) between breeds.

**Table 2 animals-14-01163-t002:** Summary statistics of the SNP genotypes in the CYP2E1 gene used in this study.

SNP	Location from ATG Codon	Allele Counts			MAF	Mutation
		0	1	2		
S35	−159	49	12	4	0.154	T > C
S36	−586	20	35	10	0.423	T > C
S102	−1693	11	21	33	0.331	C > T
S34	−1806	10	35	20	0.423	C > T
S81	−2322	21	33	11	0.423	A > T
S90	−2369	37	23	5	0.254	T > C
S103	−2514	21	33	11	0.423	A > G

**Table 3 animals-14-01163-t003:** Allele substitution effect estimates on CYP2E1 mRNA expression (fold change).

SNP	Estimate ^1^	Std. Error	Significance Level
S35	−0.74	1.23	0.55
S102	−3.38	1.28	0.01
S90	−3.57	1.57	0.03

^1^ Estimates of fold change were calculated in a combined SNP effect model, which includes breed effect and weight of the pigs as linear covariates.

## Data Availability

Data are contained within the article.

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
