# Peer review of "Effect of SNPs in the Promoter Region on the Expression of Cytochrome P450 2E1 (CYP2E1) in Pig Liver"

_animals, 2024, doi:10.3390/ani14081163_

Round 1

Reviewer 1 Report

Comments and Suggestions for Authors

Submitted for review is a manuscript entitled: Effect of SNPs in the promoter region on the expression of Cytochrome P450 2E1 (CYP2E1) in pig liver. The authors have not avoided some errors and shortcomings. I have indicated my suggestions or questions/concerns below:

Note 1: Page 1-2; Section introduction - This section definitely lacks elaboration on pig breeding and husbandry and makes reference to the subject of the manuscript. Hence, information on pig breeds that are characterised by a highly odorous smell should be highlighted here.

Note 2: Page 2; Section introduction - The authors here are citing the work of Whittington et al. [10] however, it should be specified here which breed has been shown to have low hepatic expression of this gene (i.e. Meishan X Large White pigs). The same is true below with the Squires and Lundström citation [7]. The authors should revise the manuscript further and detail this data in both the introduction and discussion.

Critical Note 3: Page 2; Section Material and methods, subsection: Animals - It is at this section that the authors should specify the number of individual breeds that were involved in the study. This information can be found later in this section. Furthermore, here it is very important to state the age of the animals tested. Especially since in this type of study the age factor can have a significant impact. In addition, information on the environmental and nutritional conditions under which the animals were kept, as well as information on the exact method of decapitation, should be included in this section. Hence, the question that is raised here is whether the authors of this study should not have the approval of the relevant Ethics Committee.  If not - the regulations and legislation under which the authors conducted the experiment should be included in section MM. If yes - add the number of the relevant Ethics Committee approval.

Note 4: Page 2; Section Material and methods, subsection: SNP Discovery- It should be clear on what basis boars were divided into those with high and low levels of taint.

Critical Note 5: Page 4; Section Results, subsection: CYP2E1 mRNA and Protein Expression A figure of the Western blot results should be added here.

Author Response

Comments and Suggestions for Authors

Reviewer 1

Submitted for review is a manuscript entitled: Effect of SNPs in the promoter region on the expression of Cytochrome P450 2E1 (CYP2E1) in pig liver. The authors have not avoided some errors and shortcomings. I have indicated my suggestions or questions/concerns below:

Note 1: Page 1-2; Section introduction - This section definitely lacks elaboration on pig breeding and husbandry and makes reference to the subject of the manuscript. Hence, information on pig breeds that are characterised by a highly odorous smell should be highlighted here.

Authors: We have added information on the differences in boar taint among different breeds of pigs and included a new reference [4] that summarizes this information.

Note 2: Page 2; Section introduction - The authors here are citing the work of Whittington et al. [10] however, it should be specified here which breed has been shown to have low hepatic expression of this gene (i.e. Meishan X Large White pigs). The same is true below with the Squires and Lundström citation [7]. The authors should revise the manuscript further and detail this data in both the introduction and discussion.

Authors: We have added this information on the variation in CYP2E1 expression and skatole levels in different breeds. This point is also emphasized in the discussion.

Critical Note 3: Page 2; Section Material and methods, subsection: Animals - It is at this section that the authors should specify the number of individual breeds that were involved in the study. This information can be found later in this section. Furthermore, here it is very important to state the age of the animals tested. Especially since in this type of study the age factor can have a significant impact. In addition, information on the environmental and nutritional conditions under which the animals were kept, as well as information on the exact method of decapitation, should be included in this section. Hence, the question that is raised here is whether the authors of this study should not have the approval of the relevant Ethics Committee.  If not - the regulations and legislation under which the authors conducted the experiment should be included in section MM. If yes - add the number of the relevant Ethics Committee approval.

Authors: We have added the number of animals in each breed and that they were sourced from commercial swine breeders in Ontario Canada and raised under commercial pig production conditions. They were slaughtered at the University of Guelph Meat lab at the weight and age range listed in the text. The study was approved by the Canadian Council of Animal Care and the University of Guelph Animal Care Committee.

We also changed the order between section 2.1 and 2.2, to follow the logical order of having SNP discovery first and then genotyping and expression analyses. This should also make clear that the animals for SNP discovery and for expression analyses were different.

Note 4: Page 2; Section Material and methods, subsection: SNP Discovery- It should be clear on what basis boars were divided into those with high and low levels of taint.

Authors: As included and highlighted in the text, 10 boars with high boar taint and 10 boars with low boar taint from each breed were used for SNP discovery. The animals were selected on the threshold values of 0.5 ug/g fat for androstenone and 0.25 ug/g fat for skatole. The objective here was to select animals at the extreme ends of the boar taint phenotype from each breed so that any relevant SNPs could be found in the promoter of the CYP2E1 gene.

Critical Note 5: Page 4; Section Results, subsection: CYP2E1 mRNA and Protein Expression A figure of the Western blot results should be added here.

Authors: We have included a figure that shows that CYP2E1 mRNA (from qPCR) and CYP2E1 protein (from Western blotting) were not correlated in any of the breeds, since we feel that this the most important result for this data.

Reviewer 2 Report

Comments and Suggestions for Authors

In this manuscript the authors identify several SNPs in the promoter region of Cytochrome P450 2E1 (CYP2E1) trough sequencing using pools of genomic DNA and determine their potential effect on the expression of CYP2E1 mRNA and protein in order to understand the genetic basis for differences in metabolism of boar taint compounds. 

The article has few minor deficiencies: 

I suggest the authors to attach the following as additional material to this article: images of electrophoresis gels, sequencing results.

Please do not leave space between the value and Celsius degrees (°C) and between value and percent (%). Please be consistent throughout the manuscript.

I recommend the manuscript to be considered for publication after minor revision. 

Regarding In the citations, from the total of 32 citation, 10 are self-citations by authors.

Author Response

Comments and Suggestions for Authors

Reviewer 2

In this manuscript the authors identify several SNPs in the promoter region of Cytochrome P450 2E1 (CYP2E1) trough sequencing using pools of genomic DNA and determine their potential effect on the expression of CYP2E1 mRNA and protein in order to understand the genetic basis for differences in metabolism of boar taint compounds. 

The article has few minor deficiencies: 

I suggest the authors to attach the following as additional material to this article: images of electrophoresis gels, sequencing results.

Authors: We did not include images of electrophoresis gels, since we have already published this before (Weircinska et al., 2012; ref [17] in the revised manuscript) and have highlighted this in the text. We obtained our sequencing results from a commercial sequencing service at the University Health Network, Toronto so we feel it is not necessary to show images of these results since this is now a routine procedure.

Please do not leave space between the value and Celsius degrees (°C) and between value and percent (%). Please be consistent throughout the manuscript.

Authors: We have made these edits.

I recommend the manuscript to be considered for publication after minor revision. 

Regarding In the citations, from the total of 32 citation, 10 are self-citations by authors.

Authors: This issue of self-citation was also pointed out by the technical editor. We have removed 4 of the 10 references and here is the justification for keeping these references:

3,   this is a recent review article from our group that covers details of the synthesis and degradation of boar taint compounds.

7, this paper shows the negative relationship between CYP2E1 and skatole and its metabolites.

9,  we have removed this reference.

13  this paper describes the FImpute method for imputation of genotype sequences.

14  this paper describes the methods for qPCR that we used previously.

16, this paper describes the methods for Western blotting that we used previously.

17  we have removed this reference.

18  this paper shows the effects of CYB5 on CYP2E1 activity and also has an image of the western blot for CYP2E1.

23  we have removed this reference.

32  we have removed this reference.

Reviewer 3 Report

Comments and Suggestions for Authors

The manuscript “Effect of SNPs in the promoter region on the expression of Cytochrome P450 2E1 (CYP2E1) in pig liver” written by Archer and colleagues contain some controversial points. First, authors discussed reducing of Boar taint and role of Cytochrome P450 2E1 in its regulation.  Next, SNP in promoter region of were associated with CYP2E1 gene expression, but no direct connection between expression level of CYP2E1 mRNA and protein level of Cytochrome P450 2E1 protein. Thus, there no connection between promoter located SNP in CYP2E1 and boar taint could be shown. From other side some interesting points could be found in population related part of this study. I would recommend authors to completely revise manuscript and reduce discussion of boar taint in favor to population study.  

Major points

Statistical analysis of transcription data looks not adequate: Pearson correlation coefficient must be applied to the data distributed by the Normal distribution. However, transcription data not distributed normally, thus rank (Spearman) correlation is necessary to apply. Additionally, significance (p-value) not giving any tendencies, it could only support significance (p.4, last paragraph). These points should be corrected.     

Author Response

Comments and Suggestions for Authors

Reviewer 3

The manuscript “Effect of SNPs in the promoter region on the expression of Cytochrome P450 2E1 (CYP2E1) in pig liver” written by Archer and colleagues contain some controversial points. First, authors discussed reducing of Boar taint and role of Cytochrome P450 2E1 in its regulation.  Next, SNP in promoter region of were associated with CYP2E1 gene expression, but no direct connection between expression level of CYP2E1 mRNA and protein level of Cytochrome P450 2E1 protein. Thus, there no connection between promoter located SNP in CYP2E1 and boar taint could be shown. From other side some interesting points could be found in population related part of this study. I would recommend authors to completely revise manuscript and reduce discussion of boar taint in favor to population study. 

Authors:  CYP2E1 is an enzyme that is important in the metabolism of skatole, a major component of boar taint. Therefore, we feel that it is important to describe this functional role of CYP2E1 in boar taint metabolism, and we have added additional information on this point in response to Reviewer #1. However, the manuscript does focus on the role of promoter SNPs in regulating the expression of CYP2E1 and we found that two of these SNPs are associated with CYP2E1 mRNA expression, but not with CYP2E1 protein expression. This novel finding suggests that factors that regulate the translation of CYP2E1 mRNA into protein may also be important in regulating the metabolism of skatole by CYP2E1.

We have also added a simple summary that emphasizes the focus of this manuscript on the SNPs that regulate expression of CYP2E1.

Major points
Statistical analysis of transcription data looks not adequate: Pearson correlation coefficient must
be applied to the data distributed by the Normal distribution. However, transcription data not
distributed normally, thus rank (Spearman) correlation is necessary to apply. 

Authors: Pearson correlation was not used with the transcription data, but rather a linear regression, which does not need to assume normality of the response variable (CYP2E1 protein expression in our analyses) and the explanatory variables are assumed as known and fixed (CYP2E1 mRNA fold change in our analyses). So, to calculate the regression coefficient normality is not required. For testing the regression coefficient, then normality of the response variable (CYP2E1 protein expression in our analyses) is required (in fact the normality of the residuals of the model are required). Even though the F-tests from the ANOVA are known to be quite robust to deviation from normality, we tested the normality of CYP2E1 protein expression and the residuals of the regression model and null hypothesis of normality was not rejected (P>0.05) for any of the three breeds for both the CYP2E1 protein expression and the residuals of the regression model.

The text in the material and methods and results were changed to clarify that a regression analysis was used to test the association between CYP2E1 protein expression and the CYP2E1 mRNA fold change and that CYP2E1 protein expression followed a normal distribution within and across breeds. 

Additionally, significance (p-value) not giving any tendencies, it could only support significance (p.4, last paragraph). 

Authors: The text was changed to address your comment.

Round 2

Reviewer 1 Report

Comments and Suggestions for Authors

The authors analyzed the reviewer's comments and responded to them in the review. Moreover, all required changes were made to the text of the manuscript. Therefore, I believe that the manuscript in its current form is suitable for publication. 

Author Response

Thank you for your comments that the manuscript is now ready for publication

Reviewer 3 Report

Comments and Suggestions for Authors

In the corrected version of manuscript “Effect of SNPs in the promoter region on the expression of    Cytochrome P450 2E1 (CYP2E1) in pig liver” authors made it more balanced and I think, that it could be published in the Animals. However, I wonder, why authors didn’t use statistical analysis more traditional way. In biology linear regression is not very frequent, that’s why several other methods are used for uncover potential connection. I would recommend test relation between CYP2E1 RNA and protein by the Spearman or Kendall correlation, because rank correlation is free from linear relationships and could uncover non-linear connections.  

Author Response

Thank you for your suggestion. We have run the suggested Spearman and Kendall correlation between CYP2E1 RNA fold change and protein expression to uncover possible nonlinear  biological  relationships; however no significant positive correlation was found both across and within breeds. In addition, we re-ran the linear regression analyses including quadratic and cubic terms and again the nonlinear terms were not significant. These results agree with the dispersion of data points depicted in Figure 1.